# Stylize and Align: Unlabeled-Image Stylized Continuous Consistency Regularization for Hand Pose Estimation in the Wild

## Abstract

Hand pose estimation has become a cornerstone of advanced human behavior understanding. In particular, 3D hand pose estimation has seen significant attention, with numerous approaches being proposed. However, it is unclear whether the modern approaches are applicable to real-world scenarios directly. We are focused on the robustness of hand pose estimators in the wild, noting that existing datasets exhibit distinct differences from real-world data. Thus, despite great advances, there remains considerable room for improvement, as most recent efforts have primarily focused on model architectures or on datasets within limited environments. To this end, we present a novel approach that unifies two key techniques: style transfer using unlabeled in-the-wild images to enhance data diversity (*i.e.*, Stylize) and continuous consistency regularization (CCR) to capture fine-grained relations between hand pose data, providing rich supervisory signals (*i.e.*, Align). To evaluate the robustness of the learned representations through our framework, we demonstrate that our method significantly enhances generalization capabilities across various tasks, including 3D hand pose estimation and transfer learning for 2D hand pose estimation, all within our designed real-world testbed. Notably, these improvements are achieved using less than 5% of the data size compared to a large-scale dataset, InterHand2.6M.

## 1 Introduction

Hand pose estimation tasks, particularly in 3D, have gained increasing attention across various fields, such as motion capture, human-computer interaction, augmented reality, and virtual reality. This task focuses on reconstructing a single person's right hand in 2D/3D space. Recent studies on single-hand pose estimation, which is the main focus of this paper, can be broadly categorized into two classes: refining model architectures and generating datasets.

Recently, reconstructing a single hand from monocular RGB images (Cai et al., 2018; Zimmermann & Brox, 2017) has become the de facto standard in the field. There are two primary approaches: model-based and model-free. Model-based approaches (Kanazawa et al., 2018; Moon et al., 2022a; Park et al., 2022) use a pre-defined parametric model (*i.e.*, MANO (Romero et al., 2017)) by forwarding their predicted MANO parameters (*i.e.*, pose and shape) to MANO layers for hand reconstruction. On the other hand, model-free approaches (Kolotouros et al., 2019; Choi et al., 2020) directly reconstruct the 3D hand from an input image without a parametric model. To improve accuracy, both approaches have increasingly adopted advanced architectures, including transformer (Park et al., 2022; Lin et al., 2021b;a) or graph convolutional network (Ge et al., 2019; Tang et al., 2021; Lin et al., 2021a; Li et al., 2022), going beyond traditional convolutional neural networks. Although they have been proven to be effective, there is still room for further improvement in terms of task-specific regularization which can simply serve as an add-on to existing methods.

As another direction, the research community has spent significant effort in collecting 3D hand datasets. One of the seminal datasets for the markerless capture of 3D hand pose is FreiHAND (Zimmermann et al., 2019), which employs a multi-view camera setup to capture various hand poses with the use of a green screen. Recently, several datasets designed to address specific challenges (*e.g.*, hand-object interaction (Hasson et al., 2019; Hampali et al., 2020; Chao et al., 2021) and

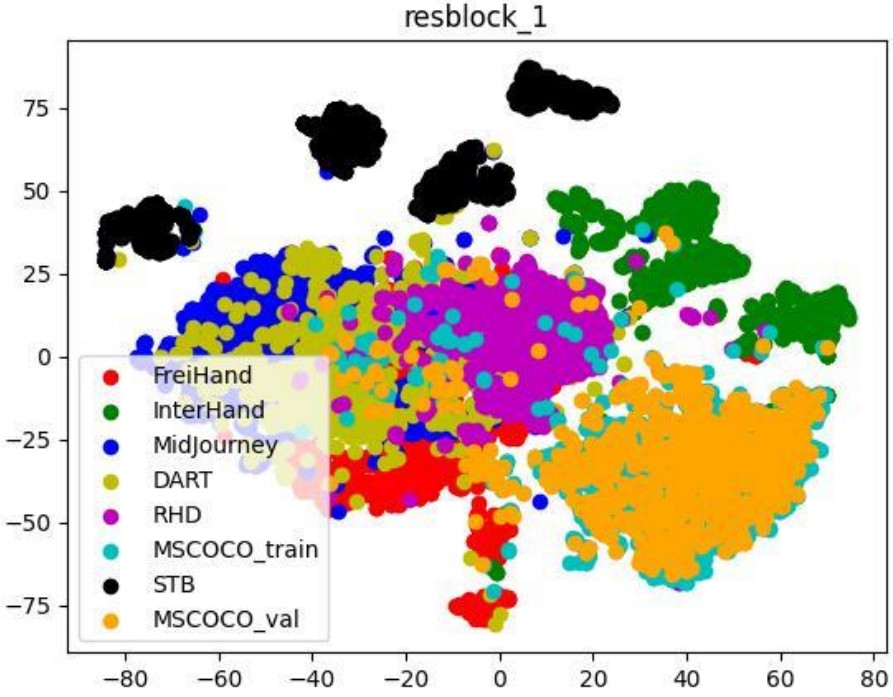

Figure 1: t-SNE visualization of the style statistics (concatenation of mean and standard deviation) computed from the first residual block's feature maps of a ResNet, known as style descriptors. The visualization clearly shows that Lab/synthetic datasets differ significantly from in-the-wild images in terms of style and appearance.

blurred hand (Oh et al., 2023), in 3D hand pose estimation have been proposed. Notably, Inter-Hand2.6M (Moon et al., 2020) has been proposed to offer a large-scale collection of accurate 3D hand pose data, including diverse poses from single-hand gestures to interacting hand scenarios. However, since these laboratory datasets (Lab datasets) are generated in controlled studio environments, they have limited stylistic variations (*e.g.*, colors and backgrounds), which are far from those of in-the-wild images. A straightforward way to resolve this issue is to collect a large-scale 3D hand dataset composed of in-the-wild images and corresponding 3D ground truths (GTs). However, it is highly demanding, as capturing 3D data requires numerous calibrated, synchronized cameras, making it labor-intensive to set up in diverse outdoor locations.

This paper is motivated by the observation: the significant visual discrepancy between Lab datasets and in-the-wild images, as illustrated in Fig. 1. To this end, we propose a novel framework that unifies current dominant techniques: style transfer (*i.e.*, Stylize) and consistency regularization (*i.e.*, Align) to close the gap between monotonous Lab datasets and complicated real-world environments. Specifically, we leverage the unlabeled real-world images (*e.g.*, Flickr and ImageNet (Deng et al., 2009)) as style references, injecting their individual styles into training images (*e.g.*, FreiHAND) on-the-fly during training. By utilizing easily accessible unlabeled data, our method efficiently transfers real-world knowledge into the model, allowing it to experience data with diverse styles while preserving accurate 3D GTs. Next, inspired by the success of metric learning in various areas, our method incorporates the metric learning approach to align the differently stylized training images using a relaxed consistency regularization based on continuous 3D pose GTs. This continuous consistency regularization allows the model to learn fine-grained similarities and disparities between 3D poses, providing richer supervisory signals that go beyond merely matching individual 3D pose GTs.

We demonstrate the efficacy of our framework in 3D hand pose estimation for real-world scenarios. Since this protocol has been relatively underexplored, we implement a testbed that simulates the target scenario for evaluation. Moreover, since our approach can be applied to various tasks, we also show that our method can enhance the capability of the transfer learning. Notably, our framework achieves significant improvements while using less than 5% of the data size compared to the model trained on the large-scale dataset, InterHand2.6M.

## 2 RELATED WORK

**RGB-based single hand reconstruction.** Significant strides in pose estimation have made RGB-based methods the standard in the field. Existing approaches can be categorized into model-based and model-free classes. An elementary example of the model-based approach is HMR (Kanazawa et al., 2018), which predicts parameters for a predefined hand model (*i.e.*, MANO (Romero et al., 2017)) to achieve hand reconstruction. HMR operates as an end-to-end framework, incorporating adversarial loss to ensure anatomically realistic results. On the other hand, model-free approaches bypass parametric models entirely, directly estimating 3D mesh vertex coordinates. Recent approaches in this category have employed advanced architectures like transformers (Lin et al., 2021b) and graph convolutional networks (Lin et al., 2021a), setting new benchmarks in performance. Concurrently, the community has focused on generating accurate datasets. FreiHAND (Zimmermann et al., 2019) introduced a dataset capturing single-hand poses and meshes using a portable multi-camera setup, featuring green-screen backgrounds and various composited scenes. Additionally, specialized datasets targeting challenges such as hand-object interaction (Hasson et al., 2019; Hampali et al., 2020; Chao et al., 2021) and blurred hands (Oh et al., 2023) in 3D hand pose estimation have been introduced. Remarkably, InterHand2.6M (Moon et al., 2020) provides the first large-scale, real-captured dataset with accurate 3D ground truths for both single and interacting hands. Despite these advancements, there remains a gap in the applicability of these approaches to real-world scenarios. In this paper, we introduce a novel framework that leverages simple yet effective techniques, tailored for in-the-wild applications, without the need for complex training or costly annotations.

**Neural style transfer.** The foundational work by Gatys et al. (2016) demonstrated that the style of an image can be effectively captured using the Gram matrix of a feature map within a neural network. Building upon this, Johnson et al. (2016) extended this idea, enabling the transfer of neural styles to arbitrary images. Further advancements by Dumoulin et al. (2017); Huang & Belongie (2017) revealed that style information is preserved within the lower layers of convolutional neural networks (CNNs) through instance-level feature statistics. To harness this, Huang & Belongie (2017) introduced Adaptive Instance Normalization (AdaIN), a technique that replaces the scale and shift parameters with feature statistics derived from an external input, thus facilitating arbitrary style transfer. In a different vein, recent studies, such as Geirhos et al. (2019), have uncovered that CNNs exhibit a strong bias toward style information. This observation has led to a surge of interest in leveraging neural style transfer for visual recognition tasks. From a data augmentation perspective, MixStyle (Zhou et al., 2021) introduces a method that perturbs style information by interpolating the scale and shift parameters of randomly paired images within a mini-batch. Conversely, UniStyle (Lee et al., 2022) seeks to de-stylize input images by applying zero-mean standardization to intermediate feature maps during both training and inference. Moreover, the Style-agnostic Network (Nam et al., 2021) utilizes adversarial training to disentangle style and content, encouraging the model to focus more on the content information. Our method is also motivated by recent studies that regularize CNN training through neural transfer via AdaIN, but with the distinct purpose of efficiently distilling in-the-wild style knowledge from readily accessible images.

**Consistency regularization.** Consistency regularization (Sajjadi et al., 2016; Laine & Aila, 2017; Zhai et al., 2019) is a widely used technique in semi-supervised learning (SSL) for image data. The core idea is to ensure that the model remains stable when an unlabeled example is augmented in ways that preserve its semantics. Therefore, data augmentation plays a crucial role in consistency regularization. Berthelot et al. (2019); Sohn et al. (2020) leverage both consistency regularization and data augmentation, establishing state-of-the-art performance in SSL image classification. Additionally, in the field of generative modeling, Zhang et al. (2020) enforce the discriminator to remain invariant under data augmentation, thereby focusing more on semantic and structural changes between real and fake data. However, the aforementioned approaches rely on binary supervision (*i.e.*, whether pairs share the same label or not). This poses significant challenges when adapting these

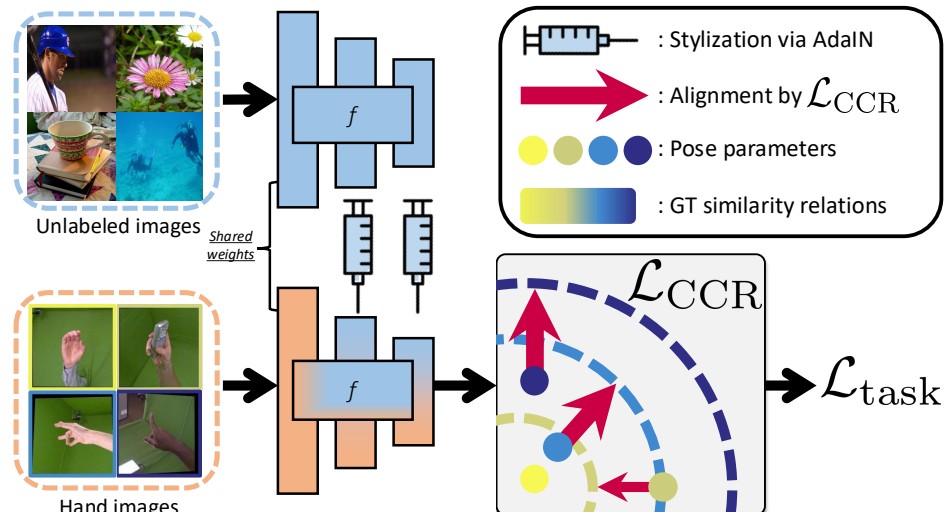

Figure 2: Illustration of the overall architecture of our framework..

methods to tasks involving continuous labels (*e.g.*, hand pose estimation). Meanwhile, the metric learning community has developed advanced methods to relax this constraint. For example, Kim et al. (2019) introduced a log-ratio loss, a variant of triplet loss, which preserves the ratios of distances between continuous labels in the learned metric space, enabling the model to capture the degree of similarity. Building on this, Zheng et al. (2020) enhances the log-ratio loss by introducing a dense structural loss that not only exploits the relationships among triplets but also incorporates all possible quadruplets within a mini-batch. Our method adopts a form of relaxed consistency regularization as a supervised learning, distinct from SSL. In detail, we directly apply this technique to predicted 3D poses based on 3D pose GTs to provide the model with rich supervisory signals. These signals capture fine-grained similarities and differences between various 3D poses, aiming at prevention of overfitting (*e.g.*, memorization) and enhancing robustness in the wild.

## 3 METHOD

As shown in Fig. 2, our framework consists of two steps: Stylization and Alignment. The former step uses adaptive instance normalization (AdaIN) (Huang & Belongie, 2017) to enhance data diversity by transferring styles from unlabeled real-world images to training hand images. The latter step employs continuous consistency regularization (CCR) to offer richer supervisory signals, capturing fine-grained relations among pose data. Details of each step are given in the following sections.

### 3.1 STYLIZATION: ADAIN WITH UNLABELED IMAGES

**Review of style transfer via AdaIN.** The goal of style transfer is to blend the visual style of a source image with the content of a target image, which results in a new image that reflects the source's aesthetic characteristics while retaining the target's structural elements. Recent studies (Ulyanov et al., 2016; Dumoulin et al., 2017; Huang & Belongie, 2017) have shown that normalizing feature tensors using instance-specific mean and standard deviation is effective in removing the style of an image, a technique commonly referred to as Instance Normalization (IN). Specifically, let $F \in \mathbb{R}^{C \times H \times W}$ denote an intermediate feature map of an image $x$. IN can be formulated as:

$$\text{IN}(F) = \gamma \frac{F - \mu(F)}{\sigma(F)} + \beta, \tag{1}$$

where $\gamma, \beta \in \mathbb{R}^C$ are learnable affine transformation parameters, and $\mu(F), \sigma(F) \in \mathbb{R}^C$ are the channel-wise mean and standard deviation, defined as:

$$\mu_c(F) = \frac{1}{HW} \sum_{h=1}^{H} \sum_{w=1}^{W} F_{c,h,w}, \tag{2}$$

and

$$\sigma_c(F) = \sqrt{\frac{1}{HW}\sum_{h=1}^{H}\sum_{w=1}^{W}(F_{c,h,w} - \mu_c(F))^2}, \tag{3}$$

where $\mu(F) = [\mu_1(F),\ldots,\mu_C(F)]$ and $\sigma(F) = [\sigma_1(F),\ldots,\sigma_C(F)]$. Finally, Huang & Belongie (2017) introduced adaptive instance normalization (AdaIN), which replaces the scale and shift parameters in Eq. (1) with the feature statistics of another intermediate feature map (*i.e.*, $F_s$) of the style image (*i.e.*, $x_s$) to achieve arbitrary style transfer:

$$\text{AdaIN}(F, F_s) = \sigma(F_s)\frac{F - \mu(F)}{\sigma(F)} + \mu(F_s), \tag{4}$$

**Hand stylization via unlabeled in-the-wild images.** In contrast to conventional style transfer work which attaches a decoder for image generation, our approach aims to expose the model to diverse style information of real-world images without involving any decoder or image synthesis process. Namely, we propose a content-aware stylization that transfers the styles of additional unlabeled in-the-wild images to training hand images via AdaIN. This is based on our core intuition that unlabeled in-the-wild images can provide the model with valuable knowledge of real-world visual styles.

Thus, given an in-the-wild image $x_i$ from an external source (*e.g.*, Flickr, ImageNet, web-crawled images), we stylize the training hand image $x_h$ using the following operation:

$$\text{Stylize}(F_h, F_i) = \sigma(F_i)\frac{F_h - \mu(F_h)}{\sigma(F_h)} + \mu(F_i), \tag{5}$$

where $F_h$ and $F_i$ represent the intermediate feature maps of $x_h$ and $x_i$, respectively. In practice, in-the-wild images are randomly sampled from their source and then each is matched to a single training hand image sampled from a specific dataset (*e.g.*, FreiHAND) in an instance-wise manner. By default, our proposed stylization is applied to the outputs of the 1st and 2nd residual blocks, as we have empirically found it effective when applied to multiple early layers. Notably, we do not use any labels from the in-the-wild images, even if the dataset provides them.

## 3.2 ALIGNMENT: CCR BETWEEN HAND POSE DATA

**Review of consistency regularization.** Consistency regularization (CR) has become a fundamental component of recent state-of-the-art semi-supervised learning algorithms (Berthelot et al., 2019; Sohn et al., 2020). A common strategy in this approach is data augmentation, where input transformations are applied under the assumption that they do not alter the original discrete semantics (*e.g.*, dog or cat). The key idea is to enforce model predictions to remain consistent across these valid data augmentations, which adds the regularization term to be optimized as

$$D(x, Aug(x)) = \|f(x) - f(Aug(x))\|_2^2, \tag{6}$$

where $x$ represents an arbitrary image, $f$ is the mapping function from the image space to output representation, and $Aug$ refers to a stochastic data augmentation.

**Continuous consistency regularization.** While CR has been highly successful, it is not directly applicable to tasks with continuous labels (*e.g.*, 3D hand pose estimation where the pose labels are 48-dimensional) since it relies on binary labels (*i.e.*, whether the pair shares the same label). For instance, enforcing CR between an anchor data point and other samples with different GT poses—comprising the majority of the dataset—is infeasible, leaving significant room for further improvement. Motivated by this, we propose to introduce continuous consistency regularization (CCR) tailored for hand pose estimation from a metric learning perspective. The core idea is to pull or push a pair of samples in the hand pose space according to their GT pose distance.

More specifically, inspired by recent studies in the metric learning community (Kim et al., 2019; Zheng et al., 2020) that focus on preserving relative distances between samples in the embedding space, our method incorporates this approach into a CCR loss term defined as:

$$\mathcal{L}_{\text{CCR}}(a, i, j) = \left(\log\frac{D(f(x_a), f(x_i))}{D(f(x_a), f(x_j))} - \log\frac{D(y_a, y_i)}{D(y_a, y_j)}\right)^2, \tag{7}$$

where $(a, i, j)$ are the indices of a triplet, with $a$ as the anchor and $i, j$ as its neighbors (*i.e.*, $(x_a, x_i, x_j)$ are the triplet images, and $(y_a, y_i, y_j)$ are the corresponding hand pose GTs). The function $f$ maps the image space to the hand pose space (*i.e.*, $f(x)$ is a 48-dimensional hand pose prediction), and $D(\cdot)$ denotes the squared Euclidean distance. This loss, a variant of the triplet loss without positive-negative separation, enables the model to learn a metric that reflects the hand pose distance between data pairs. Consequently, incorporating this regularization allows the model to capture continuous pose relationships more effectively than using only the standard task loss.

Finally, the overall objective of our end-to-end framework combines $\mathcal{L}_{\text{CCR}}$ with the standard loss functions for the target task (e.g., minimizing errors in predicted MANO parameters and 3D joint coordinates for 3D hand pose estimation) as follows:

$$\min \mathcal{L}_{\text{total}} = \mathcal{L}_{\text{task}} + \lambda \mathcal{L}_{\text{CCR}}, \qquad (8)$$

where $\mathcal{L}_{\text{task}}$ is the standard task loss, and $\lambda$ balances the contribution of $\mathcal{L}_{\text{CCR}}$. Note that we adopt the sampling strategy from Kim et al. (2019) to enhance $\mathcal{L}_{\text{CCR}}$. For details, please refer to the Appendix.

## 4 EXPERIMENTS

In this section, we evaluate the effectiveness of the proposed framework across 3D single-hand pose estimation, and transfer learning for 2D pose estimation. These evaluations are commonly conducted within our custom-designed testbed, which is specifically tailored for accurate assessment in real-world scenarios.

We start with the implementation details, covering the architecture and baseline datasets used in all experiments, followed by both quantitative and qualitative results on the aforementioned tasks.

### 4.1 BASELINE ARCHITECTURE

Among the various model architectures available, we selected SHNet, a model that is widely adopted in the pose estimation community for both hand-related (Moon, 2023; Moon et al., 2024) and body-related (Moon et al., 2022a;c) studies. Our choice was further motivated by the compatibility of SHNet with our method, allowing us to seamlessly integrate our proposed components—stylization and continuous consistency regularization—into its architecture. Specifically, these components are applied to the early layers (*i.e.*, the first and second ResBlocks) and the pose output space of SHNet, all without requiring any additional modifications to the existing structure. For more details of their implementation, please refer to the Appendix.

### 4.2 DATASETS

**Baseline datasets.** For our experiments, we established the baselines using existing datasets, specifically FreiHAND (Zimmermann et al., 2019), HO3D (Hampali et al., 2020), and Inter-Hand2.6M (Moon et al., 2020). Notably, for InterHand2.6M, we focused exclusively on single-hand data, utilizing the right-hand data with its ground truths (GTs) and augmenting it by horizontally flipping the left-hand data to create additional right-hand examples with corresponding GTs. This resulted in a total of 687,547 samples in our experimental results for InterHand2.6M.

**Test dataset.** To evaluate the robustness in real-world scenarios, we used the MSCOCO (Lin et al., 2014; Jin et al., 2020) as our test set. MSCOCO offers a comprehensive collection of images from a wide range of natural, everyday scenes, accompanied by rich ground truths (GTs) for various tasks, including hand keypoints. Additionally, a recent study (Moon, 2023) provided MANO GTs for the whole-body version of the MSCOCO dataset using NeuralAnnot (Moon et al., 2022b) for training purposes. Although these MANO GTs were generated for training in Moon (2023), we utilized this dataset exclusively as a test set in our experiments, ensuring that no model had prior access to it. We believe that this dataset best simulates in-the-wild conditions with highly accurate 3D hand annotations. Similar to our approach with InterHand2.6M in our experiments, we focused exclusively on single-hand data, resulting in a total of 26,851 samples for evaluation.

**Unlabeled dataset for our stylization.** Among various possible options, following existing approaches that use external data to improve model generalization (Yue et al., 2019; Chen et al., 2020b;

Table 1: Performance comparison of SHNet trained on various 3D hand datasets, with all results evaluated on the 3D-labeled MSCOCO single-hand dataset for real-world applications. †: Only the green-screen background portion of FreiHAND was used, which comprises 1/4 of the total dataset.

| Settings | #data↓ | PA-MPJPE↓ | PA-MPVPE↓ |
|---|---|---|---|
| FreiHAND | 0.13M | 15.29 | 15.06 |
| HO3D | 0.08M | 13.75 | 14.07 |
| FreiHAND+HO3D | 0.21M | 13.47 | 13.60 |
| InterHand2.6M | 0.68M | 14.57 | 14.38 |
| Ours on FreiHAND | 0.13M | **12.23** | **12.38** |
| Ours on FreiHAND† | **0.03M** | 12.54 | 12.68 |

Table 2: 2D hand pose estimation performance of linear heads on the MSCOCO validation dataset, trained on representations learned with different pretraining settings. †: The setting of the used data size is the same as in Table. 1

| Pretraining setups | #data↓ | PCK↑ | EPE↓ |
|---|---|---|---|
| Random Init | 0 | 71.82 | 53.00 |
| ImageNet | 1.2M | 77.62 | 48.05 |
| FreiHAND | 0.13M | 77.83 | 47.83 |
| HO3D | 0.08M | 78.04 | 47.84 |
| FreiHAND+HO3D | 0.21M | 78.62 | 47.02 |
| InterHand | 0.68M | 77.28 | 47.86 |
| Ours on FreiHAND† | **0.03M** | **80.23** | **44.84** |

Huang et al., 2021), we adopt ImageNet (Deng et al., 2009) as the unlabeled dataset for stylizing hand images unless stated otherwise. ImageNet, with millions of images across thousands of categories, offers diverse visual examples, making it suitable for our method. Although not specifically designed for hand pose estimation, its scale and variety effectively support our stylization process.

### 4.3 3D HAND POSE ESTIMATION

**Setups.** In this experiment, we integrate the MANO layer into our framework for 3D single-hand reconstruction, as SHNet employs a model-based approach. Specifically, the MANO layer reconstructs the 3D hand based on the predicted MANO parameters (*i.e.*, pose and shape) from by SHNet. For evaluation, we utilize two commonly adopted metrics: PA-MPJPE (Procrustes-Aligned Mean Per Joint Position Error) and PA-MPVPE (Procrustes-Aligned Mean Per Vertex Position Error).

**Results.** As summarized in Table. 1, we observe that our method outperforms all the models learned with the Lab datasets even the least training data. the model trained on InterHand2.6M fails to generalize effectively to in-the-wild images, despite its substantial data size. This outcome substantiates our assertion that Lab datasets, despite their scale, exhibit clear limitations in their ability to generalize to unseen data.

### 4.4 TRANSFER LEARNING FOR 2D HAND POSE ESTIMATION

**Setups.** To assess the quality of the representations learned through our framework, we conduct transfer learning experiments on 2D hand pose estimation, following the widely adopted linear evaluation protocol (Chen et al., 2020a; He et al., 2020). In this approach, a linear head for 2d hand pose estimation is trained on top of the frozen representations obtained during pretraining. In the first stage, we train all models using their respective pretraining setups based on contrastive learning (Chen et al., 2020a), except for our method, which replaces contrastive learning with our proposed continuous consistency (CCR) regularization. In the second stage, we train only the linear heads on the MSCOCO training dataset, while keeping the pretrained representations frozen. We then evaluate the performance of the linear heads on the MSCOCO validation dataset. We use the Percentage of Correct Keypoints (PCK) and End-Point Error (EPE) as the evaluation metrics to gauge the performance of the 2D hand pose estimation task.

**Results.** As summarized in Table 2, our method consistently outperforms all models trained with baseline setups, even when using the least amount of training data. Interestingly, although ImageNet pretraining is primarily designed for general image classification tasks, unrelated to hand pose es-

Table 3: Ablation study of the proposed components on 3D hand pose estimation on FreiHAND.

| Methods | PA-MPJPE | PA-MPVPE |
|---|---|---|
| FreiHand | 15.29 | 15.06 |
| w/ Stylization | 12.75 | 12.86 |
| w/ CCR | 13.02 | 13.13 |
| Ours | **12.23** | **12.38** |

Table 4: Ablation study on the effect of unlabeled images for our proposed stylization in 3D hand pose estimation on FreiHAND.

| Stylization | PA-MPJPE | PA-MPVPE |
|---|---|---|
| None | 15.29 | 15.06 |
| ImageNet20K | 12.86 | 13.00 |
| ADE20K | 13.26 | 13.39 |
| BDD20K | **12.81** | **12.90** |

timation, its learned representations outperform those of InterHand2.6M-pretrained models. This can be attributed to the broad real-world knowledge encapsulated in ImageNet-pretrained representations, which enables better generalization to real-world data, regardless of the specific pretraining task. This finding reinforces our claim that incorporating visual in-the-wild stylization during training is crucial, and our proposed CCR further enhances the ability of the trained model to generalize to diverse, real-world environments.

### 4.5 IN-DEPTH ANALYSIS

**Ablation study.** We conducted an ablation study to assess the individual contributions of each component in our framework, as summarized in Table 4. The results demonstrate that both stylization and continuous consistency regularization (CCR) contribute to improved generalization. Notably, the stylization component shows a larger reduction in error, highlighting its effectiveness in enhancing the capability of the learned model to generalize to diverse, real-world data. Lastly, these components complement each other, significantly boosting performance.

**Impact of types of unlabeled images.** To evaluate the effect of different types of unlabeled images on our proposed stylization, we trained models using various datasets with identical data sizes (*i.e.*, 20K) for a fair comparison. As shown in Table 3, stylization with in-the-wild datasets consistently enhances model performance. This suggests that incorporating diverse visual styles, even from datasets not specifically designed for hand pose tasks, improves generalization. However, exploring which specific characteristics of these datasets lead to the most effective stylization remains an open question, which we leave for future work.

## 5 CONCLUSION

We introduced a framework combining in-the-wild stylization via AdaIN and continuous consistency regularization (CCR) to improve the generalization of hand pose estimation models. Our approach enhances the model's robustness using diverse, real-world styles and fine-grained 3D pose alignment, outperforming existing methods with less data. The results highlight the limitations of lab datasets and the importance of real-world data in improving model generalization.

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

# A    APPENDIX

## A.1    IMPLEMENTATION DETAILS

**Sampling strategy for $\mathcal{L}_{\textbf{CCR}}$.** To efficiently explore diverse triplets, we employ dense triplet sampling as proposed by Kim et al. (2019). In this approach, we combine all pairs of neighbors with the anchor, while excluding duplicate triplets where the order of neighbors does not impact $\mathcal{L}_{\text{CCR}}$. Specifically, for each anchor, we select its $k$ nearest neighbors based on pose distance, with additional neighbors randomly sampled from the remaining dataset. The search space for $k$ is defined as $\{\lfloor (B-1)/2 \rfloor, B-1\}$, where $B$ is the batch size. Note that the same number of training steps is used across all experiments to ensure fair comparisons with our method.

**3D hand pose estimation.** We use ResNet-50 as the backbone, following the original SHNet (Moon et al., 2022a; Moon, 2023). The hyperparameters include a batch size of 64 and 100 epochs.

$$\text{nDCG}_K(q) = \frac{1}{Z_K} \sum_{i=1}^{K} \frac{2^{r_i}}{\log_2(i+1)}, \tag{9}$$

where $K$ represents the number of top retrievals considered, and $Z_K$ is a normalization constant ensuring that $\text{nDCG}_K$ has a maximum value of 1. The relevance score $r_i$ is defined as $r_i = -\log_2\left(\|y_q - y_i\|_2 + 1\right)$, which decreases logarithmically with the Euclidean distance between the query $q$ and the $i$th retrieval. The score is further discounted by $\log_2(i + 1)$ to give higher importance to top-ranked retrievals. A higher nDCG indicates better retrieval quality.

**Transfer learning for 2d hand pose estimation.** For evaluation metrics, we report Percentage of Correct Keypoints (PCK) (higher is better) and End-Point Error (EPE) (lower is better). For the architecture, we use ResNet-18 as the backbone and an MLP for the linear head, which is attached to the backbone. For the hyperparameters in each setting, we use a batch size of 512 and 100 epochs for both the pretraining stage (*i.e.*, the first stage) and the linear evaluation stage (*i.e.*, the second stage).

