# OpenReview forum: "Stylize and Align: Unlabeled-Image Stylized Continuous Consistency Regularization for Hand Pose Estimation in the Wild"
_ICLR.cc/2025/Conference — ICLR 2025 Conference Withdrawn Submission_

### Official Review · Reviewer_EMBG · 2024-10-28

**Soundness:** 2
**Presentation:** 2
**Contribution:** 2
**Rating:** 3
**Confidence:** 4

**Summary:**

This paper tackles a challenge in 3D hand pose estimation: The existing datasets are captured in laboratory settings and have a distinct appearance from real-world settings due to the high annotation costs for 3D hand pose. To address the gap between the training datasets in laboratory settings and the real-world test data, this paper proposes two techniques: style transfer-based feature augmentation and continuous consistency regularization (CCR). The former augments the features extracted from hand images to various styles using AdaIN layers and real-world images such as the ImageNet dataset. The latter aligns the distance between the predicted poses and the distance between the GT labels. The experimental results show the effectiveness of the proposed method.

**Strengths:**

The strengths of this paper are as follows:

i) The motivation for tackling the 3D hand pose estimation and its challenge (i.e., the gap between laboratory and real-world settings) is well-discussed in Sec. 1.

ii) The concrete procedure of the proposed method and the idea behind the proposed method are clearly described in Sec. 3.

iii) The performance of the proposed method is good. In addition, the efficacy of each component of the proposed method is confirmed in the ablation study.

**Weaknesses:**

The weaknesses of this paper are as follows:

i) My main concern is about a lack of technical contributions. As described Sec. 2, the data augmentation using style transfer is a common approach and not novel. In Sec. 2, it is stated that "Our method is also motivated by recent studies that regularize CNN training through neural transfer via AdaIN, but with the distinct purpose of efficiently distilling in-the-wild style knowledge from readily accessible images". However, the only technical difference from MixStyle is the use of external datasets, such as ImageNet, as the source-style images. In addition, the CCR is completely the same as the log-ratio loss. It should be clarified which part of the proposed method is technically novel.

ii) I do not agree with the discussion about consistency regularization (CR) in Sec. 3.2. Specifically, it is stated that "While CR has been highly successful, it is not directly applicable to tasks with continuous labels since it relies on binary labels". In the case of CR between different images that have the same category label, this is true. However, in the case of CR between the original input image and its augmented image as written in Eq. (6), this is not true, and we can optimize Eq. (6) because the two images have the same continuous label. Even if the augmentation changes the GT labels, we can perform CR by estimating the transformation of the GT labels as [A] and [B] did. The discussion in Sec. 3.2 should be carefully revised because this discussion is the core motivation for introducing the log-ratio loss.

[A] Yang et al., "SemiHand: Semi-supervised Hand Pose Estimation with Consistency", ICCV, 2021.

[B] Ohkawa et al., "Domain Adaptive Hand Keypoint and Pixel Localization in the Wild", ECCV, 2022

iii) The proposed method is evaluated only on the MSCOCO dataset. Although other datasets with 3D hand pose GT may not be in real-world settings, the evaluation on various test datasets can support validating the model generalization ability from the proposed method.

**Questions:**

N/A

---

### Official Review · Reviewer_MCF1 · 2024-11-03

**Soundness:** 2
**Presentation:** 3
**Contribution:** 2
**Rating:** 5
**Confidence:** 4

**Summary:**

This paper addresses a crucial yet unresolved issue in hand pose estimation: the domain gap between annotated datasets from lab environments and in-the-wild data. This gap leads to poor performance of models trained on lab datasets when applied to real-world
scenarios. The authors attribute this domain gap primarily to the visual differences between real-world images and those captured in the lab, proposing style transfer as a solution. They adopt AdaIN in existing pose estimation frameworks to transfer in-the-wild image knowledge into the network. Additionally, they proposed continuous consistency regularization to maintain consistency in pose predictions across images with domain gaps.

**Strengths:**

- The domain gap problem explored in this paper is highly relevant to the field of hand pose estimation, and the use of style transfer and consistency regularization presents potential applicability across a wide range of existing methods.

**Weaknesses:**

- Vector graphics should be used.
- The experimental setup and evaluation scheme in this paper are insufficiently robust, making it difficult to assess whether the model trained with style transfer and consistency regularization can generalize to in-the-wild datasets.
  -  When evaluating the robustness of the model trained with style transfer and CCR in real-world scenarios, only FreiHAND contain in-the-wild data. It is still necessary to investigate the impact of more in-the-wild dataset on the method's performance like [1, 2].
  - The test sample is obtained from the MSCOCO dataset. What is the specific strategy for sampling? Is this sufficient to cover the diversity of real-world scenarios? The authors should provide clarification on the diversity of the test samples, such as hand pose, hand shape, and view.
- How does the consistency regularization approach address the visual discrepancy between lab datasets and in-the-wild images? The authors should rewrite the motivation for introducing consistency regularization.
- Fig. 2's illustration of CCR seems less clear than the formulas presented in the paper, which offer a more intuitive understanding of the concept. The authors should redraw a clearer illustration of CCR.

[1] Reconstructing Hands in 3D with Transformers. CVPR 2024.

[2] WiLoR: End-to-end 3D Hand Localization and Reconstruction in-the-wild. ArXiv 2024.

**Questions:**

- In this paper, only SHNet was used as the baseline model. More models such as HRNet and ViT should be introduced to test the generalizability of the proposed approach.

---

### Official Review · Reviewer_1nAb · 2024-11-03

**Soundness:** 2
**Presentation:** 2
**Contribution:** 2
**Rating:** 3
**Confidence:** 5

**Summary:**

This work presents a method for hand pose estimation in the wild, using stylization to capture realistic visual features and continuous consistency regularization (CCR) to align similar hands together in 3D space. The stylization is done using AdaIN from ImageNet features whereas CCR helps to correlate distances in pose and feature space. Experiments on in-the-wild MSCOCO images with models trained on FreiHand, HO3D, and InterHand show benefits of the proposed modifications.

**Strengths:**

- Most of the evaluations are done on the in-the-wild MSCOCO images, going beyond the controlled datasets.
- The idea to correlate 3D pose and feature space of hands using a consistency loss is intuitive.
- Experiments on MSCOCO (Tab.1,2) show the effectiveness of the proposed components in both 3D pose estimation (Tab.1) and linear probing for 2D keypoints (Tab.2) tasks.
- The benefits are more prominent when working in the low dataset regime (Tab.1,2).
- Ablations (Tab.3,4) show benefits from both the proposed components.

**Weaknesses:**

- A common way [A,B,C] to deal with the lack of visual diversity in individual lab datasets, is to train across several datasets. Recent works [A,B,C] have used this strategy to show good performance in the wild. It is unclear if stylization would still help when training on multiple datasets as in these works [A,B,C]. Since code is available for these works along with the datasets processed into a common format (e.g. codebase of HaMeR [A]), it'd be useful to conduct this experiment. One aspect where stylization might be useful is data efficiency for training. Currently, the results on FreiHand, HO3D & InterHand seem to indicate this, but training across multiple datasets might be even more effective.
- It'd be useful to have details on data augmentations used during training. Existing works [A,B,D] use image-level augmentations to improve the visual diversity (e.g. color scale in [A] or pixel noise in [D]). Is stylization more effective than these augmentations? Currently, this is not mentioned in the paper. A comparison to these augmentations is needed to verify the benefits of stylization. Another way of doing augmentation is to replace the green screen in FreiHand with different backgrounds from ImageNet or internet images, this would also increase the visual diversity. These experiments should also be conducted to check if stylization via AdaIN is indeed beneficial.
- Some clarifications are required about implementation details:
    - During stylization, is the hand also stylized or just the background (using hand masks)? If the hand is stylized, then the ground truth pose might also change. Is this taken into account?
    - It'd also be useful to check if the AdaIN is indeed helping in stylization. This can be done by training a decoder to generate the stylized image. This will also help to verify that hands are not getting affected.
    - For applying CCR, is the relative hand pose used or the absolute hand pose? L272 mentions a 48-dimensional pose, is this in MANO format? If yes, this would include the relative hand pose and the global orientation of the hand? Depending on the coordinate system used for the global hand orientation, the distances can vary. It'd be useful to clarify this.
    - The 48-dimensional pose consists of rotations between different joints and the global orientation of the hand, which are represented in axis angle format. Is there any reason why the Euclidean distance metric (L272-273 and Eq.7) is used to measure the distance between rotations? This may lead to issues, e.g. as in [E], and alternate representations [E] or metrics (e.g. angular errors) might be more suitable.
    - In Eq.7, how is $y$ represented? Is it the 3D locations of the hand keypoints or MANO parameters? L272-273 mentioned that $f(x)$ is 48-dimensional, is this the MANO parameters?
- A straightforward way to measure the 2D keypoint error is to project the 3D keypoints in the image and measure the distance to the ground truth 2D keypoints. Is there any specific reason why this is not done in the evaluation? Instead, linear probing is used (Sec. 4.4).
- It'd also be useful to have results on absolute hand poses (i.e. MPJPE) and also visualizations for qualitative comparisons.

[A] Pavlakos et al. Reconstructing Hands in 3D with Transformers. CVPR 2024
[B] Prakash et al. 3D Hand Pose Estimation in Everyday Egocentric Images. ECCV 2024
[C] Rong et al. FrankMocap: A Monocular 3D Whole-Body Pose Estimation System via Regression and Integration. ICCV 2021 Workshops
[D] Fan et al. ARCTIC: A Dataset for Dexterous Bimanual Hand-Object Manipulation. CVPR 2023
[E] Zhou et al. On the Continuity of Rotation Representations in Neural Networks. CVPR 2019

**Questions:**

There are 3 major concerns (details above):
- It is unclear if stylization is indeed more useful than common strategies like image augmentations in existing works (e.g. color scale in [A] or pixel noise in [D]) and datasets (e.g. using green screen with hand masks in FreiHand to augment the background with ImageNet samples).
- Since several works use a combination of datasets for visual diversity to improve generalization, it is not evident if stylization would provide benefits over multi-dataset training (e.g. HaMeR [A] codebase already provides multiple datasets processed into a common format).
- Several details about the implementation and experiments are missing.

---

### Note · Authors · 2024-11-14

**Comment:**

We sincerely thank all the reviewers for the constructive comments to improve our paper. We decided to withdraw our paper, yet will try our best to reflect all of the valuable comments in our next manuscript.

**Withdrawal Confirmation:**

I have read and agree with the venue's withdrawal policy on behalf of myself and my co-authors.